# Comparative Study of Lipid Profile for Mice Treated with Cyclophosphamide by HPLC-HRMS and Bioinformatics

**DOI:** 10.3390/metabo15010060

**Published:** 2025-01-16

**Authors:** Ekaterina Demicheva, Fernando Jonathan Polanco Espino, Pavel Vedeneev, Vadim Shevyrin, Aleksey Buhler, Elena Mukhlynina, Olga Berdiugina, Alicia del Carmen Mondragon, Alberto Cepeda Sáez, Aroa Lopez-Santamarina, Alejandra Cardelle-Cobas, Olga Solovyova, Irina Danilova, Jose Manuel Miranda, Elena Kovaleva

**Affiliations:** 1Institute of Natural Sciences and Mathematics, Ural Federal University, Ekaterinburg 620075, Russia; fpolankoespino@urfu.me (F.J.P.E.); zellist@mail.ru (A.B.); elena.mukhlynina@bionica.science (E.M.); soloveva.olga@urfu.ru (O.S.); i.g.danilova@urfu.ru (I.D.); 2Institute of Immunology and Physiology of the Ural Branch of the Russian Academy of Sciences, Ekaterinburg 620049, Russia; o.berdyugina@iip.uran.ru; 3Institute of Chemical Engineering, Ural Federal University, Ekaterinburg 620002, Russia; p.a.vedeneev@urfu.ru (P.V.); v.a.shevyrin@urfu.ru (V.S.); e.g.kovaleva@urfu.ru (E.K.); 4Laboratorio de Higiene, Inspección y Control de Alimentos, Departamento de Química Analítica, Nutrición y Bromatología, Campus Terra, Universidade da Santiago de Compostela, 27002 Lugo, Spain; alicia.mondragon@usc.es (A.d.C.M.); alberto.cepeda@usc.es (A.C.S.); aroa.lopez.santamarina@usc.es (A.L.-S.); alejandra.cardelle@usc.es (A.C.-C.); josemanuel.miranda@usc.es (J.M.M.)

**Keywords:** lipidomics, mass spectrometry, biomarkers, blood

## Abstract

**Purpose:** Immunodeficiency conditions, which are characterized by reduced immune activity that promotes the development of chronic diseases, are needed for efficient monitoring. A promising area of monitoring and early diagnosis of immunodeficiency diseases is the determination of metabolic biomarkers in the blood. **Methods:** In this work, we identified a set of lipid biomarkers of immunodeficiency states by performing high-performance liquid chromatography–high-resolution mass spectrometry (HPLC-HRMS) analysis of blood plasma samples from mice and processing them with bioinformatics approaches. Potential biomarkers were selected through statistical analysis and further validated by MS/MS. **Conclusions:** As a result, 15 lipids were confirmed and selected as potential biomarkers of immunodeficiency states. The selected biomarkers can be further studied and serve as promising targets for the early diagnosis of immunodeficiency diseases.

## 1. Introduction

Secondary or acquired immunodeficiencies (IDs) are a group of conditions of various etiologies characterized by a disruption of the immune response mechanisms, which leads to a decrease in the functional activity of the immune system [1]. Immunodeficiency not only is an independent disease but also develops under the influence of physical and chemical factors, insufficient protein nutrition, impaired neurohumoural regulation, and oncological diseases, including long-term drug therapy. It is also believed that secondary immunodeficiency can be caused by stress, pesticides, alcohol or tobacco abuse, and drugs. IDs are widespread and are associated with reduced resistance to the effects of viral and bacterial infections [2].

Currently, a promising direction for monitoring the course of diseases is the determination of specific metabolite markers (biomarkers) in plasma. A set of these markers at specific concentrations is linked to the type and severity of the disease, making it possible to identify associations between disease states and a particular set of biomarkers at certain concentrations. The aforementioned approach is rapidly evolving due to the emergence of metabolomics, which can be defined as the study of small molecules in biofluids accompanied by bioinformatic methods [3]. Although a wide range of metabolites are studied by metabolomics, lipids receive particular interest. To date, more than 13,000 articles dedicated to lipidomics have been published in PubMed, which is much more common than in other metabolomics areas. Lipids have gained special attention because of their vital functions in organisms: they are part of all cell membranes, they perform the main energy function, and their derivatives serve as vital signaling molecules [4]. In addition, the increased number of some lipid classes in plasma, such as glycerophospholipids and polyketides, may indicate the presence of pathogenic microbiota. For example, polyketides are known to be produced by microorganisms during their life cycle [5]. These observations allow us to draw conclusions about the interconnections between the studied diseases and the microbiome.

One of the existing models for studying immunodeficiency *in vivo* involves inducing laboratory animals with cyclophosphamide (CPA) [6]. This medication is commonly used for cancer treatment; moreover, it has been effectively applied in modeling immune disorders. CPA acts by causing immunosuppression in organisms via the inhibition of regulatory T cells, which allows the use of this drug in modeling the immunodeficient state.

In this work, we explore the lipidomic profile of CPA-induced immunodeficiency in mice and identify novel plasma biomarkers of the immunodeficient state via high-performance liquid chromatography with high-resolution mass spectrometry (HPLC-HRMS) and bioinformatics methods. To assess lipid composition, we conducted three experiments each on 10 blood samples, of which 5 were collected from healthy mice and the other 5 from mice with experimentally induced immunodeficient state. The three experiments were conducted over the course of a year to assess the repeatability and reproducibility of the method and the results.

## 2. Materials and Methods

### 2.1. Animals

All manipulations with animals were performed in accordance with Directive 2010/63/EU of the European Parliament and of the Council on the protection of animals and were approved by the Ethics Committee of the Institute of Immunology and Physiology of the Ural Branch of the Russian Academy of Sciences (IIP UB RAS) (Protocol № 03-24). In this study, 30 three-month-old male C3H mice were selected as experimental subjects. The animals were maintained under standard laboratory conditions in the vivarium of the IIP UB RAS. All animals were housed at a constant temperature of 20–22 °C and a relative humidity of 55% under a 12 h light/dark cycle and were provided with food and water *ad libitum*.

### 2.2. Experimental Model of Immunodeficient State

Cyclophosphamide was used to induce the experimental model of immunodeficiency. All the mice were randomly divided into two groups of fifteen animals, namely, the control group and the immunosuppressed group. The mice were administered a single intraperitoneal injection of cyclophosphamide (Endoxan^®^, Baxter Oncology GmbH, Halle (Westfalen), Germany) at a dose of 200 mg/kg body weight. The control mice received a single intraperitoneal injection of 0.9% physiological saline in an equal volume. The weight of the animals was measured before exposure.

### 2.3. Blood Samples Collection and Primary Preparation

On the 7th day, all the mice were weighed and anesthetized with isoflurane before blood collection. Peripheral blood samples were collected from the femoral vein into clean tubes with tripotassium EDTA (K3-EDTA). After blood collection, the mice were sacrificed by dislocation of the cervical spine. Whole blood without cooling to a volume of 0.5 mL was used for hematologic analysis and flow cytometry. Another part of the blood was used for HPLC-MRMS. The blood samples were centrifuged at 1000× *g* for 15 min at 4 °C. The obtained plasma was divided into 100 μL aliquots and stored at −80 °C until HPLC-MRMS was performed. Femoral bone samples were used to obtain and study bone marrow.

### 2.4. Immunosuppression Model Verification

#### 2.4.1. Hematologic Parameters

Hematologic analysis was carried out via a Mindray BC-2800Vet Automated Hematology Analyzer (Mindray, Shenzhen, China) to measure the 18 parameters presented in Table 1.

#### 2.4.2. Flow Cytometry

Flow cytometry was conducted via a Cytomics FC500 flow cytometer (Beckman Coulter, Brea, CA, USA), with a CXP Cytometer 3.0 software (Beckman Coulter, Brea, CA, USA). The instrument was calibrated by evaluating a standardized fluorescent particle flow-check fluorosphere kit (Beckman Coulter, Brea, CA, USA). Monoclonal antibodies from Invitrogen (Waltham, MA, USA) and BioLegend (Carlsbad, CA, USA) were used to stain blood samples. Sample preparation and staining were performed according to the manufacturer’s protocol. The following fluorescent dyes were used: FITC, PE, PC5, and PC7. The following clusters of differentiation were detected via four-color analysis: CD45, CD3, CD19, CD49, CD4, CD8, CD5, and CD27. Red blood cell lysis was carried out via no-wash technology with OptiLyse C reagents (Beckman Coulter, Brea, CA, USA). Next, hematologic blood analysis was performed to determine the absolute lymphocyte content required to convert flow cytometry data into absolute values.

#### 2.4.3. Statistical Analysis

The immunosuppression data analysis was performed with the statistical software package STATISTICA v.6.0 (StatSoft Inc., Tulsa, OK, USA). The data are presented as the arithmetic mean (M) ± standard error of the mean (m). The nonparametric Wilcoxon test was used to measure the weight of animals before and after treatment, and the Mann–Whitney U test was used to test the hypothesis of homogeneity of two independent samples. A 5% significance level was used for testing statistical hypotheses.

### 2.5. Biomarkers Identification

#### 2.5.1. Lipid Extraction

Lipid extraction was carried out on each plasma sample using the methyl tert-butyl ether (MTBE) which, according to the literature, allows the extraction of the most common lipid classes from plasma [7]. Aliquots of 100 μL thawed plasma were mixed with 300 μL of methanol in a 2.0 mL centrifuge tube, stirred and shaken at room temperature for 10 min. Next, 1000 μL of MTBE was added to the resulting mixture and stirred (vortexed) for 1 h at room temperature. Afterwards, 250 μL of water was added to the mixture and stirred (vortexed) for 10 min followed by centrifugation (1360× *g*, 10 min at 15 °C). The upper organic phase was transferred to another centrifuge tube. The organic layer was evaporated to dryness using nitrogen current at 30 °C. The dry residue was reconstituted in 300 μL of chloroform–methanol mixture (1 : 9 *v*/*v*).

#### 2.5.2. HPLC-HRMS Analysis

The prepared plasma samples were analyzed using an Agilent 1290 Infinity II high-performance liquid chromatograph and an Agilent 6545 Q-TOF LC/MS tandem quadrupole time-of-flight mass spectrometer (Agilent Technologies, Santa Clara, CA, USA). The mass spectrometer was coupled with an electrospray ionization source in positive and negative ion modes. Nitrogen was used as a drying gas, with a flow rate of 10 L/min at 350 °C. The superheating gas temperature was set at 350 °C with a flow rate of 11 L/min. The other parameters were set as follows: fragmentor voltage, 120 V; cone voltage, 1000 V; capillary voltage, 3500 V. The scanning range was set to 100–1700 *m/z* in MS mode and 30–1500 *m*/*z* in MS/MS mode, with a scanning speed of 2 spectra/s. Spectra in MS/MS mode were obtained from collision-induced dissociation with nitrogen molecules at collision energies of 20, 30 and 40 eV.

Chromatographic separation was carried out on a Zorbax Eclipse Plus C18 column (100 mm × 2.1 mm × 1.8 μm), which was additionally protected by a precolumn (5 mm × 2.1 mm × 1.8 μm). In positive ion mode, a mixture consisting of 60% (vol.) acetonitrile and 40% (vol.) aqueous solution of 10 mM ammonium formate with the addition of 0.1% (vol.) formic acid (solvent A) and a mixture of 90% (vol.) isopropanol and 10% (vol.) acetonitrile with the addition of 0.1% (vol.) formic acid and a solution of 10 mM ammonium formate in 5 mL of water (solvent B) was used as the mobile phase components. In the negative ion mode, a mixture consisting of 60% (vol.) acetonitrile and 40% (vol.) aqueous solution of 10 mM ammonium acetate (solvent A) and a mixture of 90% (vol.) isopropanol and 10% (vol.) acetonitrile with a solution of 10 mM ammonium acetate in 5 mL of water (solvent B) were used as components of the mobile phase. Chromatographic separation was carried out in gradient mode via the following parameters: a linear change in the eluent B content of the mobile phase from 15 to 30 vol% for 3 min, then to 48% by 3.7 min, then to 72% by 12.7 min, and finally to 99% by 17.2 min. After that, the final eluent B content was held for up to 21 min. The flow rate of the mobile phase was 0.4 mL/min.

The temperature of the column thermostat was set to 50 °C. The prepared solution was injected in the amount of 1 μL in (+) or 2 μL in (−) mode of ion registration. The parameters were adjusted automatically via standard calibration solutions recommended by the instrument manufacturer.

#### 2.5.3. Data Preprocessing

The raw data files obtained from the HPLC—HRMS analysis were converted to mzML format with the ProteoWizard program. They were subsequently loaded into XCMS, an R programming language library for MS data preprocessing [8]. The preprocessing was divided into six steps as suggested by the XCMS guide [8]: data compression, peak detection, peak correction, chromatogram alignment, peak grouping, and filling in missing values. The IPO library for R was used to calculate the optimal XCMS parameters for our data [9]. For peak detection, the ObiWarp algorithm [10] was selected and configured to search for peaks with lengths of 10–30 s and a mass range of 14 ppm as specified by the IPO results. Subsequent peak correction for small peaks was performed with a window of 4 s via the MergeNeighboringPeaksParam function of XCMS. Afterwards, a chromatogram alignment was performed across the spectra of all the mice via the “adjustRtime” method, with the parameters calculated via IPO optimization. Finally, the peak grouping and filling of missing values were performed automatically via the built-in library functions groupChromPeaks and fillChromPeaks of XCMS. The following chromatograms show the lipid spectra for the negative (Figure 1) and positive (Figure 2) ion modes from the first experiment.

A summary table containing all extracted peaks was generated as the preprocessing output. The table contains 8 parameters, including peak *m/z*, rt, and the intensity values of each lipid on each sample. This summary table was used for further annotation and biomarker selection via statistical analysis.

#### 2.5.4. Lipid Filtering and Preliminary Annotation

After the preprocessing step, the filtering and correction of the obtained results were performed. For this purpose, we used the filters and functions presented in the LipidFinder workflow [11]. The filters were configured in the following order with the subsequent parameters. First, the Solvents_remover filter was applied with a solventMinFoldDiff of 5, to delete the noises found in blank samples (solvents). Next, the parameters mzFixedError and mzPPMError were set to 0.0005 and 16, respectively, indicating the error rates of our HPLC-HRMS. The Background_correction filter was subsequently used with an intenSignifCutOff parameter of 1000. Next, the contaminant_removal filter was applied with the default list of contaminants from LipidFinder. After that, the adduct_removal filter was employed with the parameters maxRTDiffAdjFrame and adductAddition set to 0.3 and “False”, respectively. Thereafter, the Outliers_correction filter was used with an intenOutlierCutOff of 3000. Next, the Sample_mean function was applied, followed by the rt_means_correction function with a maxRTDiffAdjFrame of 0.3, intensityStDev of 3, and means parameter set to “False”. Afterward, the Mass_reasigment function was applied. Finally, the BroadContaminant filter was used with outlierMinDiff, minNonZeroPoints, intenRSDCutOff, and rtSDCutOff set to 1, 6, 50, and 2, respectively.

Following the filtering process, the filtered lists of positive and negative ion peaks were passed through the “emulsifier” (amalgator) function, resulting in a combined table of peak intensities from both ion modes, which was used in the next step, lipid annotation. At this step, each peak was searched by *m/z* in the LIPID MAPS database [12] and the matched information about lipids was retrieved to create a table, including *m*/*z*, rt, name or bulk structure, formula, adductive ion, major class, category, and a link to its database page. It is important to clarify that this step represents the preliminary annotations based on the peak *m/z*, not final identifications, and is subject to additional confirmation.

#### 2.5.5. Selection of Biomarker Candidates

The summary table obtained after data preprocessing was used for subsequent statistical analysis. It was conducted in four steps: data normalization, data scaling, principal component analysis (PCA), and biomarker selection. Quantile normalization was chosen as the most suitable normalization method, as it shows the best results among other methods in studies from the fields of metabolomics and genomics [13]. The next step, data scaling, was performed to reduce the differences in the range between the traits of the metabolites studied. The range of intensities between lipids varies greatly, giving us some metabolites with intensity ranges in the thousands and others with intensity ranges in the millions. Thus, maximum–minimum normalization was applied to scale intensities to a comparable range. After that, the PCA method was used to reduce the data dimensionality into two components, allowing the visualization of potential differences between the groups as illustrated in Figure 3.

For biomarker selection, the Student’s *t* test and the Volcano plot visualization method were used. Volcano plot visualization is based on two parameters that are calculated for each lipid: the fold change and the p-value. The list of significant lipids was determined by setting cutoffs for both parameters, resulting in the formation of two groups of significantly changed metabolites: upregulated, metabolites with increased concentrations in the immunodeficient group; and downregulated, metabolites with decreased concentrations in the immunodeficient group. In this study, the following cutoffs were chosen: *p*-value < 0.05 and fold change > 2 or <−2. The volcano plot visualization is presented in Figure 4. The analysis was repeated for the three experiments, producing three lists of candidate biomarkers. These lists were compared, and a combined list was created that included only the biomarkers presented in all three experiments. The combined list was then subjected to MS/MS analysis for identification.

#### 2.5.6. MS/MS-Based Lipid Identification of Selected Biomarkers

Finally, the identification of the lipids selected as potential biomarkers was performed via the MS-DIAL program [14] based on the data from MS/MS experiments. The obtained fragmentation spectra were compared with the lipid spectra from the MS-DIAL program library, as well as the open mass spectral libraries: Riken Lipidomics (http://metabography.riken.jp, accessed on 27 April 2024) and MassBank of North America (http://massbank.us, accessed on 27 April 2024). The identified lipids were selected as the final biomarkers of the study; the complete list is presented in Table 2.

## 3. Results and Discussion

### 3.1. Immunosuppression Verification

To investigate lipidome modifications, a CPA-induced immunosuppression model was established in C3H mice. There was a notable decrease in body weight in both the experimental and control groups 7 days after CPA injection (Figure 5). However, CPA administration did not lead to fatality among the animals.

Hematological studies revealed a decrease in lymphocyte, red blood cell, and platelet counts 7 days after CPA administration; however, this decrease was not associated with a change in the total white blood cell count (Table 3) due to an increase in the granulocytes count. CPA administration leads to the inhibition of hematopoiesis [15]. Consequently, decreases in hemoglobin and hematocrit levels are also observed. However, erythrocyte indices (MCV, MCH, and MCHC) were not significantly changed. Myelosuppression, lymphopenia, and thrombocytopenia, as well as a reduction in the number of red blood cells, hematocrit, and hemoglobin levels, are apparently due to the cytostatic effect of CPA and are considered the common dose-limiting adverse effects of CPA treatment [16,17].

Flow cytometry analysis revealed that a single CPA injection at a dosage of 200 mg/kg body weight caused numerous perturbations in T and B lymphocytes in the peripheral blood of C3H mice up to day 7 (Table 4). A representative example of the determination of T, B, and NK cells after CPA injection is shown in Figure 6.

The percentage of peripheral blood T-lymphocytes increased by 53.5% after CPA administration. The percentage of CD8+ population increased by 68.4%, and the percentage of CD4+ cells increased by 47.6%. B lymphocytes are more sensitive to CPA treatment than T cells. A representative example of the determination of B1-, B2- and memory B cells after exposure to CPA is shown in Figure 7. Thus, 7 days after CPA administration, the number of B-lymphocytes decreased dramatically, and the percentage of B lymphocytes decreased 10-fold (Table 4), resulting in significant fluctuations in B-cell subpopulations. CPA critically affects B1- and B2-lymphocyte levels without affecting memory B cells. The number of B1 cells decreases 6.3-fold and the number of B2-cells decreases 16.8-fold. The immunosuppressive effect of CPA has been described by Mughal K.S. et al. (2024) [18], Zhou Q. et al. (2024) [19] and many others, and is usually used to model such pathological conditions. There is no effect on the number of NK cells; however, in other mice models, a decrease in the number of NK cells has been observed [20].

Consequently, the administration of CPA-induced an immunosuppressed state in C3H mice 7 days after treatment. This is characterized by a dramatic decrease in the number of B1 lymphocytes that generate natural antibodies, especially B2 cells, which are high-specificity antibody producers. T cells in C3H mice are prevalent under physiological conditions and obviously more resistant to CPA action.

### 3.2. The Effect of Cyclophosphamide on Plasma Lipidomic Profile

CPA is a strong alkylating agent that is widely used in cancer treatment and as an immunosuppressant. The mechanism of its action involves the transformation into an active form in the liver and the formation of cytotoxic metabolites, 4-hydroxycyclophosphamide and aldophosphamide [21]. The first metabolite promotes the formation of cross links in the DNA molecule, leading to cell cycle arrest and apoptosis. Moreover, the second metabolite induces oxidative stress within the cell, causing damage to key cellular molecules, such as lipids, proteins, and DNA [22]. In particular, studies have shown that liver diseases, such as hepatitis, liver cirrhosis, and cancer, develop against a background of activated redox reactions within the cell [23].

According to the results of HPLC-HRMS analysis and the MS/MS validation procedure, the number of potential biomarkers was narrowed down to 15 lipidomic compounds (Table 2). Among them were nine downregulated and six upregulated metabolites. Most of the downregulated metabolites belong to a group of glycerophosphocholines, a subclass of glycerophospholipids. Metabolites of a glycerophosphocholine subclass are typically the breakdown products of another compound, phosphatidylcholine, which is a part of the cell membrane. Considering the presence of glycerophosphocholines and the cytostatic effect of CPA, we hypothesize that CPA damages the cellular membranes, resulting in glycerophosphocholines formation. However, considering the oxidative stress induced by CPA, the glycerophosphocholines may potentially be rapidly converted to other smaller compounds and do not accumulate in the cells; thus, the overall number of these metabolites is downregulated [24] as shown in the results. These findings and hypotheses align with previous studies on CPA that confirm the ability of CPA to induce oxidative stress [25,26].

Among the upregulated metabolites, the majority belong to the triacylglycerols class. One possible explanation for their elevated levels is the oxidative stress induced by CPA. Under oxidative stress, stored fats are broken down into free fatty acids that transform into triacylglycerols in the liver, resulting in their high level in the blood plasma. After CPA treatment, unsaturated triacylglycerols were also observed in a study by Xu J. et al. (2022) [27]. Additionally, oxidative stress leads to the accumulation of oxidized lipids, for example, MDA and 4-HNE that can further disrupt cell structures, alter lipid metabolism, and provide a toxic effect that will provoke cell death mechanisms. The elevated level of MDA in rats after CPA administration was observed in a study by Mahmoud et al. (2016) [25]. Two sphingomyelins (SMs) were also elevated. SMs represent an important class of phospholipids in the cell membrane. The structural role of SMs is determined by their asymmetry, low degree of unsaturation, and ability to form many hydrogen bonds. In addition, SMs interact with cholesterol and other sterols to form SM/sterol domains in the membrane. It is assumed that SMs are regulators of cholesterol distribution in the membrane, which determines cholesterol homeostasis in the cell. Naturally, when cell membranes are damaged by the oxidation products of CPA, this class of lipids increases in the blood. Moreover, the cytotoxic effect of CPA, which induces DNA damage, has been shown to significantly modify SM concentrations [27,28].

## 4. Conclusions

In this study, we performed an untargeted lipidomic analysis using HPLC-HRMS to identify potential biomarkers for immunodeficiency in a CPA-induced mouse model. To confirm the immunodeficient state, plasma samples were subjected to flow cytometry analysis prior to lipidomic profiling. The results demonstrate the broad capability of HPLC-HRMS to detect a wide range of lipid biomarkers in mouse blood plasma. Through this lipidomic analysis, 15 distinct lipid biomarkers with either increased or decreased concentrations were identified, including glycerophosphocholines and triacylglycerols, which changed significantly in response to CPA-induced immunosuppression. In particular, downregulated glycerophosphocholines suggest significant membrane disruption, whereas upregulated triacylglycerols reflect increased oxidative stress. The complete set of 15 biomarkers can be further studied to identify the precise causes of their altered activity and later serve as a diagnostic tool for the early identification of immunodeficiency states.

## Figures and Tables

**Figure 1 metabolites-15-00060-f001:**
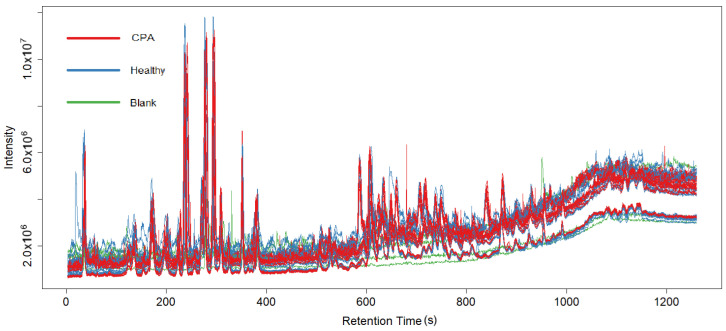
Chromatogram of lipid spectra of all plasma samples of the negative ion mode. Blue lines represent control samples, red lines represent CPA-induced immunodeficiency group, green lines represent blank sample.

**Figure 2 metabolites-15-00060-f002:**
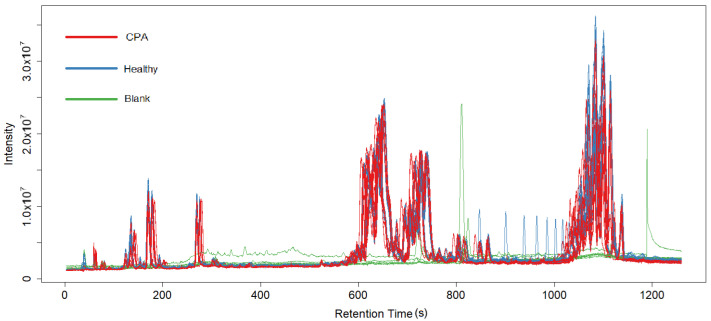
Chromatogram of lipid spectra of all plasma samples of the positive ion mode. Blue lines represent control samples, red lines represent CPA-induced immunodeficiency group, green lines represent blank sample.

**Figure 3 metabolites-15-00060-f003:**
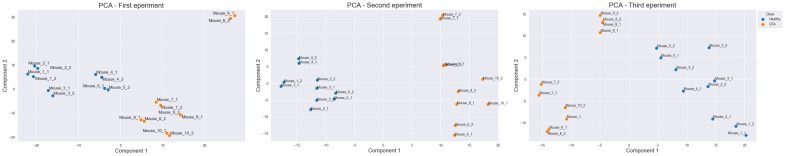
PCA plots for each of the three experiments. Blue dots represent control group, and orange dots represent the group with CPA-induced immunodeficiency.

**Figure 4 metabolites-15-00060-f004:**
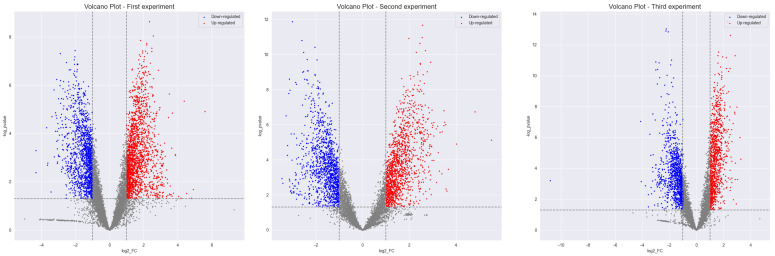
Volcano plots for each of the three experiments. Blue dots represent lipids with decreased concentration (downregulated), red dots represent lipids with increased concentration (upregulated). The selected cutoffs are *p*-value < 0.05 and fold change > 2 or <−2.

**Figure 5 metabolites-15-00060-f005:**
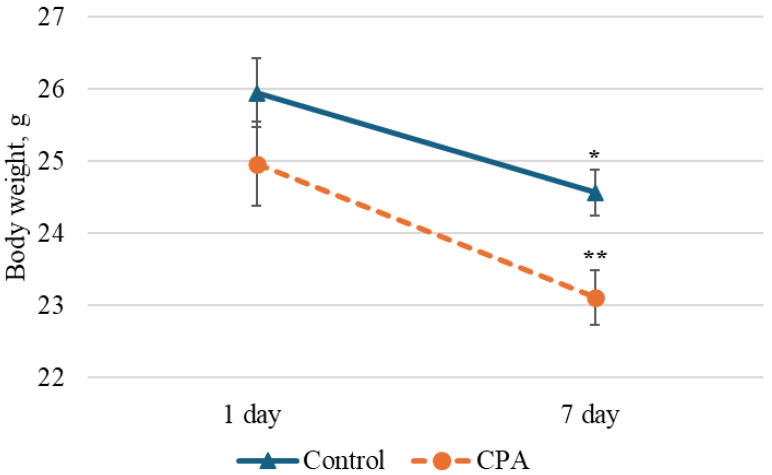
The body weight of treated mice was decreased in both groups from the first to the 7th days. ** indicates *p* < 0.0000001 compared to the first day; * indicates *p* < 0.00005 compared to the first day.

**Figure 6 metabolites-15-00060-f006:**
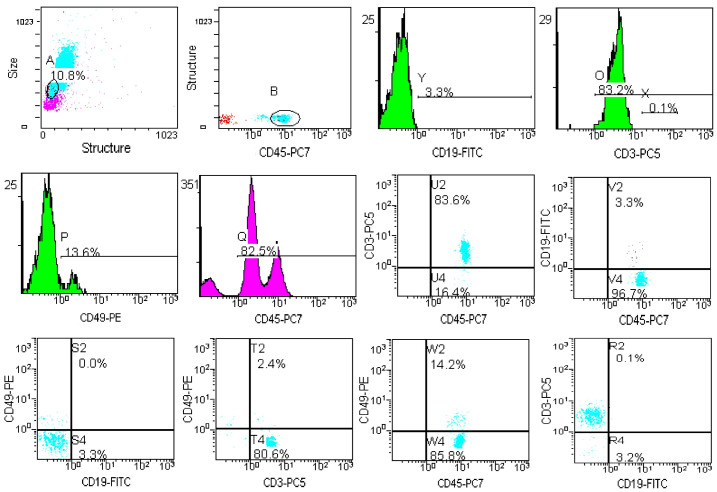
Flow cytometry plots and histograms for lymphocyte analysis after the mice exposure to CPA. Abbreviations on the x and y axes refer to different flow cytometry reagents. T-cell markers are CD45+CD3+, B cells—CD45+CD19+, NK cells—CD45+CD49+.

**Figure 7 metabolites-15-00060-f007:**
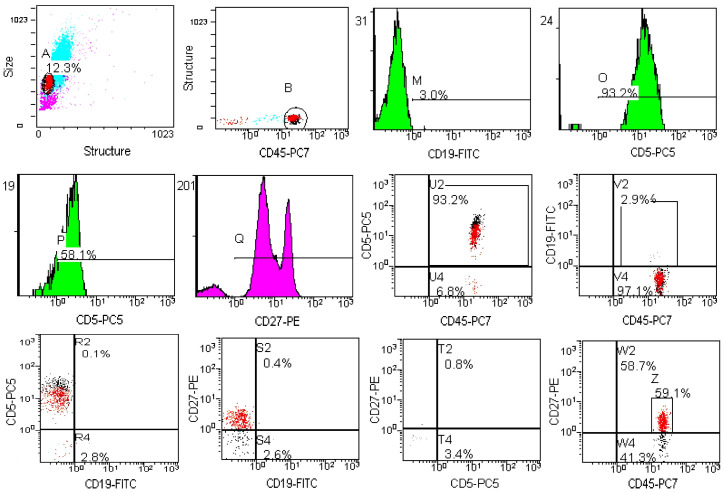
Flow cytometry plots and histograms for B-cells analysis after the mice exposure to CPA. Abbreviations on the x an y axis refer to different flow cytometry reagents. T-cell markers and B1-cell markers are CD45+CD19+CD5+CD27–, B2 cells—CD45+CD19+CD5–CD27–, memory B cells—CD45+CD19+CD5–CD27+.

**Table 1 metabolites-15-00060-t001:** Complete blood count parameters representing absolute counts (#) and percentages (%) of various cell types, along with indices describing red blood cells, hemoglobin content, and platelets.

Parameters	Description
WBC, 10^9^/L	Absolute number of leukocytes
Lym#, 10^9^/L	Absolute number of lymphocytes
Mon#, 10^9^/L	Absolute number of monocytes
Grn#, 10^9^/L	Absolute number of granulocytes
Lym%	Lymphocyte percentage
Mon%	Monocyte percentage
Grn%	Granulocyte percentage
RBC, 10^12^/L	Red blood cells
Hb g/L	Hemoglobin concentrations
Hct, %	Hematocrit
MCV, fL	Average volume of red blood cells
MCH, pg	Mean corpuscular hemoglobin content
MCHC, g/L	Mean corpuscular hemoglobin concentration
RDW, %	Red blood cell distribution width
Plt, 10^9^/L	Platelet count
MPV, fL	Mean platelet volume
PDW, %	Platelet distribution width
Pct, %	Thrombocrit

**Table 2 metabolites-15-00060-t002:** Selected potential lipidomic biomarkers tentatively identified with MS/MS experiments. The biomarkers 1–6 are upregulated, 7–15 are downregulated.

№	*t_R_*, min	Ion	Accurate Mass	Formulae	Name	Characteristic Fragment Ions	Class	*p*-Value	Fold Change
1	8.9	[M-H]^−^	883.5325	C_47_H_81_O_13_P	PI 18:1_20:4	619.2884 601.2779 597.3025 579.2923 417.2407 303.2326 297.0387 281.2489 241.0127	GPIs	$9.1220\times 10^{-4}$	4.116
2	10.2	[M+OAc]^−^	761.5807	C_39_H_79_N_2_O_6_P	SM 18:1;O2/16:0	687.5429 449.3143 168.0430 78.9590	SMs	$8.5265\times 10^{-4}$	3.356
3	14.5	[M+OAc]^−^	871.6897	C_47_H_93_N_2_O_6_P	SM 18:1;O2/24:1	797.6525 449.3136 168.0430 78.9591	SMs	$3.6798\times 10^{-4}$	2.877
4	17.2	[M+Na]^+^	1021.7247	C_67_H_98_O_6_	TG 20:4_22:6_22:6	717.4849 693.4850	TAGs	$2.2052\times 10^{-4}$	5.477
5	17.4	[M+Na]^+^	997.7272	C_65_H_98_O_6_	TG 20:4_20:4_22:6	693.4850 669.4859	TAGs	$1.2984\times 10^{-3}$	4.527
6	17.5	[M+Na]^+^	973.7271	C_63_H_98_O_6_	TG 18:2_20:4_22:6	693.4847 669.4838 645.4853	TAGs	$5.1465\times 10^{-6}$	4.870
7	3.0	[M+H]^+^	400.3419	C_23_H_45_NO_4_	CAR 16:0	85.0282	CARs	$1.6435\times 10^{-3}$	0.318
8	5.7	[M+H]^+^	552.4021	C_28_H_58_NO_7_P	LPC 20:0	534.3922 184.0726 104.1065	GPCs	$2.1956\times 10^{-4}$	0.321
9	10.7	[M+H]^+^	792.5903	C_46_H_82_NO_7_P	PC O-18:2_20:4	506.3602 184.0727	GPCs	$3.2279\times 10^{-3}$	0.310
10	11.8	[M+H]^+^	768.5915	C_44_H_82_NO_7_P	PC O-16:0_20:4	585.5215 482.3594 184.0730	GPCs	$6.6409\times 10^{-3}$	0.379
11	12.9	[M+H]^+^	720.5900	C_40_H_82_NO_7_P	PC O-16:0_16:0	537.5199 482.3588 184.0729	GPCs	$9.3740\times 10^{-8}$	0.219
12	14.5	[M+H]^+^	787.6709	C_45_H_91_N_2_O_6_P	SM 18:1;O2/22:0	184.0731	PSPLs	$2.2446\times 10^{-4}$	0.330
13	18.2	[M+NH4]^+^	822.7558	C_51_H_96_O_6_	TG 14:0_16:0_18:1	577.5188 549.4873 523.4716	TAGs	$5.8895\times 10^{-4}$	0.320
14	18.2	[M+NH4]^+^	848.7726	C_53_H_98_O_6_	TG 16:0_16:0_18:2	575.5038 551.5032	TAGs	$6.8380\times 10^{-4}$	0.330
15	18.9	[M+NH4]^+^	878.8176	C_55_H_104_O_6_	TG 16:0_18:0_18:1	605.5494 579.5340 577.5182	TAGs	$6.9316\times 10^{-3}$	0.319

Abbreviations: GPIs, glycerophosphoinositols; SMs, sphingomyelins; CARs, fatty acyl carnitines; GPCs, glycerophosphocholines; PSPLs, phosphosphingolipids(sphingomyelins); TAGs, triacylglycerols.

**Table 3 metabolites-15-00060-t003:** Complete blood count parameters of mice after CPA administration, representing absolute counts (#) and percentages (%) of various cell types, along with indices describing red blood cells, hemoglobin content, and platelets.

Parameters	Control	CPA
WBC, 10^9^/L	5.24 ± 0.41	5.13 ± 0.58
Lym#, 10^9^/L	3.41 ± 0.30	1.37 ± 0.31 *
Mon#, 10^9^/L	0.20 ± 0.02	0.26 ± 0.04
Grn#, 10^9^/L	1.63 ± 0.13	3.49 ± 0.48 *
Lym%	64.48 ± 1.57	25.85 ± 3.31 *
Mon%	3.89 ± 0.23	5.59 ± 0.35 *
Grn%	31.63 ± 1.45	68.56 ± 3.45 *
RBC, 10^12^/L	8.83 ± 0.11	7.06 ± 0.12 *
Hb g/L	144.50 ± 1.06	113.41 ± 2.33 *
Hct, %	44.20 ± 0.51	35.16 ± 0.60 *
MCV, fL	50.13 ± 0.29	49.87 ± 0.24
MCH, pg	16.33 ± 0.17	16.00 ± 0.09
MCHC, g/L	326.67 ± 2.68	321.65 ± 2.08
RDW, %	12.71 ± 0.25	11.67 ± 0.13 *
Plt, 10^9^/L	1126.33 ± 58.30	556.03 ± 33.94 *
MPV, fL	5.14 ± 0.09	6.28 ± 0.10 *
PDW, %	16.07 ± 0.07	16.65 ± 0.06 *
Pct, %	0.57 ± 0.03	0.34 ± 0.02 *

* *p* < 0.01 versus the Control group (Mann–Whitney U-test).

**Table 4 metabolites-15-00060-t004:** The content of lymphocytes of various subpopulations in C3H mice with cyclophosphamide-induced immunodeficiency.

Parameters	Control	CPA
T-lymphocytes (CD45+CD3+), %	55.767 ± 2.107	85.583 ± 3.112 *
T-lymphocytes (CD45+CD3+), 10^9^/L	1.736 ± 0.127	1.878 ± 0.256
B-lymphocytes (CD45+CD19+), %	32.600 ± 2.248	3.183 ± 0.239 *
B-lymphocytes (CD45+CD19+), 10^9^/L	1.055 ± 0.157	0.070 ± 0.011 *
NK cells (CD45+CD49+), %	9.950 ± 1.353	9.783 ± 2.897
NK cells (CD45+CD49+), 10^9^/L	0.324 ± 0.060	0.188 ± 0.044
CD8(+) T cells (CD45+CD3+CD8+), %	14.250 ± 0.456	24.000 ± 0.868 *
CD8(+) T cells (CD45+CD3+CD8+), 10^9^/L	0.445 ± 0.033	0.519 ± 0.059
CD4(+) T cells (CD45+CD3+CD4+), %	41.117 ± 1.615	60.700 ± 2.986 *
CD4(+) T cells (CD45+CD3+CD4+), 10^9^/L	1.279 ± 0.091	1.344 ± 0.203
B1-lymphocytes (CD45+CD19+CD5+CD27–), %	0.817 ± 0.183	0.217 ± 0.065 *
B1-lymphocytes (CD45+CD19+CD5+CD27–), 10^9^/L	0.025 ± 0.006	0.004 ± 0.001 *
Memory B cells, (CD45+CD19+CD5–CD27+), %	0.333 ± 0.084	0.200 ± 0.058
Memory B cells, (CD45+CD19+CD5–CD27+), 10^9^/L	0.010 ± 0.002	0.005 ± 0.002
B2-lymphocytes (CD45+CD19+CD5–CD27–), %	31.692 ± 2.221	2.742 ± 0.255 *
B2-lymphocytes (CD45+CD19+CD5–CD27–), 10^9^/L	1.027 ± 0.155	0.061 ± 0.011 *

* *p* < 0.05 versus the Control group (Mann–Whitney U-test).

## Data Availability

The raw data supporting the conclusions of this article will be made available by the authors on request. The data are not publicly available due to privacy.

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
