# Peer review of "Comparative Study of Lipid Profile for Mice Treated with Cyclophosphamide by HPLC-HRMS and Bioinformatics"

_metabolites, 2025, doi:10.3390/metabo15010060_

Round 1
Reviewer 1 Report
Comments and Suggestions for Authors
This study points out potential biomarkers for cyclophosphamide treated mice as animal model for immunodeficiency. The experiments are reasonably well conducted and described. Conclusions are acceptable in spite of being constrained to some extent by the low number of animals (5) used in each experimental group, even though three experiments were conducted with 10 animals each. This issue should be explicitly indicated and discussed in the manuscript. Results from each experiment should be compared, a concern that is not clear whether was done so or not. Perhaps, an additional comparison through PCA or PLS-DA of the combined data from the three experiments would be relevant for the conclusions.
The authors should explain why analysis of representative DNA degradation products in the blood was not conducted since this is an expected effect of CPA.
As well, indicate in the manuscript whether lipid oxidation products were found in the analysis, and, if the response is negative how can this feature be explained taking into account that one of the CPA effects is known to be generation of ROS and membrane disruption (line 278 of the manuscript).
Minor issues:
1) line 68: was instead of were
2) line 111: give the centrifuge force (g units) instead of rpm
3) line 142: 50 ºC may be a high temperature for organic solvents and some lipids. Were lower temperatures checked?
4) Figures 1 and 2 legends: why so many traces in the chromatogram if it is representative of one sample? Comparative chromatograms of samples from each group would be welcome. What does the green line trace correspond to in the figure 2?
5) lines 194-196: give references for this statement. No other normalization methods were used in this study?
6) Figure 3: mice-6 sample seems to be an outlier. Analysis with mixed samples from the three experiments should be recommendable (as indicated above).
7) Table 3: use point for decimals
8) Figures 6 & 7: examples? Why not actual data? Indicate which points correspond to every group.
9) Lines 280-282: this statement is a hypothesis or do the authors have support for it?
10) Line 283: there three TAG species upregulated and three downregulated. Maybe this is not an adequate expression though correct.
Reviewer 2 Report
Comments and Suggestions for Authors
This study is focused on the assessment of changes in mouse lipid profile after treatment with cyclophosphamide. I recommend to accept this manuscript, but after a Major Revision, because I have a few concerns that must be addressed first:
Major concerns:
1. The differences uncovered by untargeted and unlabeled lipidomics seem to have high significance, but in order to have no doubts it would be usefull if Authors provide the results of targeted MS analysis of selected lipids.
2. Please explain the reasoning why Wilcoxon test was used (2.4.3. Statistical analysis; line 100). This test is used for paired samples and Authors compare two groups of different animals.
3. Figure 3: Why only 5 mice/group were used for PCA while there were 15 mice in each group?
4. In line 204 Authors write about using Student’s t test. Did they check the normality of data distribution which would allow to use this test?
5. Figure 4: Showing volcano plot with no information regarding which lipids are significantly changed has no rational use. Please at least put the list of significantly changed lipids (with the significance) to the supplementary file.
6. Discussion is very brief. In total, only 7 studies from literature were used to discuss the results obtained by the authors.
Minor concerns:
1. Figure 1 and 2: Descriptions used in that figures are mearly visible; I do not see the reason for putting the numbers on top of these figures for example 100.0026-1698.8077;
2. Figure 6 and 7 look very sloppy and unorganised: no units on the axis; Fig 6 (top, second from right) description of x-axis is not visible; sizes of each plots are different; numbers on y-axis sometimes are vertical and some times are horizontal, their fonts are also different; abbereviations used on both figures are not explained in the figure descriptions.
3. Minor language corrections (for example Authors used “of” instead of “for” in line 2; “were” instead of “was” in line 68).
Round 2
Reviewer 2 Report
Comments and Suggestions for Authors
Thank you for responding to most of my concerns. Unfortunately, I still see some issues that can be improved:
1. Identification of lipids using MS/MS approach is not the same as targeted analysis of selected lipids. In targeted analysis selected lipids are fragmented and selected fragments of these lipids are monitored and their levels are analyzed. If the Authors performed the proper targeted analysis, please add the description of the parameters to the Materials section and add to table 4 information which fragmentation ion was monitored for the specific lipid.
2. I understand that some figures were generated using specialized software in its standard output format, but there is always possibility to correct/change the description of the axes using graphical software such as Photoshop.
3. Please insert units of the Retention Time in Figure 1 and 2.
4. Table 4: P-values cannot be “0.000”. Please insert exact values even if they are lower than 10exp-4.
